# Short-Chain Fatty Acid Receptors and Cardiovascular Function

**DOI:** 10.3390/ijms23063303

**Published:** 2022-03-18

**Authors:** Anastasios Lymperopoulos, Malka S. Suster, Jordana I. Borges

**Affiliations:** Laboratory for the Study of Neurohormonal Control of the Circulation, Department of Pharmaceutical Sciences, Nova Southeastern University College of Pharmacy, Fort Lauderdale, FL 33328, USA; ms5019@mynsu.nova.edu (M.S.S.); jb3837@mynsu.nova.edu (J.I.B.)

**Keywords:** adipose tissue, cardiovascular, FFAR2, FFAR3, GPCR, hormone secretion, neuronal, signal transduction, SCFA, sympathetic

## Abstract

Increasing experimental and clinical evidence points toward a very important role for the gut microbiome and its associated metabolism in human health and disease, including in cardiovascular disorders. Free fatty acids (FFAs) are metabolically produced and utilized as energy substrates during almost every biological process in the human body. Contrary to long- and medium-chain FFAs, which are mainly synthesized from dietary triglycerides, short-chain FFAs (SCFAs) derive from the gut microbiota-mediated fermentation of indigestible dietary fiber. Originally thought to serve only as energy sources, FFAs are now known to act as ligands for a specific group of cell surface receptors called FFA receptors (FFARs), thereby inducing intracellular signaling to exert a variety of cellular and tissue effects. All FFARs are G protein-coupled receptors (GPCRs) that play integral roles in the regulation of metabolism, immunity, inflammation, hormone/neurotransmitter secretion, etc. Four different FFAR types are known to date, with FFAR1 (formerly known as GPR40) and FFAR4 (formerly known as GPR120) mediating long- and medium-chain FFA actions, while FFAR3 (formerly GPR41) and FFAR2 (formerly GPR43) are essentially the SCFA receptors (SCFARs), responding to all SCFAs, including acetic acid, propionic acid, and butyric acid. As with various other organ systems/tissues, the important roles the SCFARs (FFAR2 and FFAR3) play in physiology and in various disorders of the cardiovascular system have been revealed over the last fifteen years. In this review, we discuss the cardiovascular implications of some key (patho)physiological functions of SCFAR signaling pathways, particularly those regulating the neurohormonal control of circulation and adipose tissue homeostasis. Wherever appropriate, we also highlight the potential of these receptors as therapeutic targets for cardiovascular disorders.

## 1. Introduction

The main class of gut microbial metabolites are short-chain fatty acids (SCFAs), mainly acetate, propionate and butyrate, which are absorbed into the systemic circulation and can act as hormone signals on target tissues/cells [1,2,3,4,5,6,7,8,9,10]. SCFAs are the cognate ligands for the free fatty acid receptors 2 and 3 (FFAR2/3), both of which belong to the G protein-coupled receptor (GPCR), or 7 transmembrane (7TM)-spanning receptor superfamily [1,2]. Both FFAR2 and FFAR3, called collectively SCFA receptors (SCFARs) from here on in this review, are present in the gut [4,11,12,13], adipose tissue [14,15], bone marrow [11], liver [16], muscle [17,18], lungs [1], brain [2,19,20], heart, and peripheral sympathetic neurons [1,21]. This tissue expression pattern of SCFARs reveals their vital role in several pathologies, such as diabetes [22,23,24,25,26], obesity and metabolic syndromes [27,28,29,30], inflammatory bowel diseases [31,32], asthma [33], gout/arthritis [34], and cardiovascular diseases, including arrhythmias, heart failure, hypertension, and myocardial infarction [21,35,36,37]. Like all GPCRs, both SCFARs reside at the plasma membrane and bind endogenous SCFAs on the extracellular side of the membrane in order to activate heterotrimeric G proteins on the intracellular side [38]. Both FFAR2 and FFAR3 share the signature 7TM-spanning helical core of every GPCR [39]. When GPCRs are activated by ligands, the alpha subunits of heterotrimeric G proteins coupled to the receptor dissociate from the βγ subunits, further affecting intracellular signaling by activating or inhibiting the target functional proteins/enzymes (effectors) [40,41,42,43,44,45,46]. Four G protein families are known, identified after their representative subunit: Gs, Gi/o, G_q/11_, and G_12/13_ [47,48]. Gαs activates and Gαi inhibits adenylyl cyclase (AC), an enzyme that synthesizes the second messenger cyclic 3′, 5′-adenosine monophosphate (cAMP) from adenosine triphosphate (ATP) [49] (Figure 1). In contrast, Gαq activates a different membrane-bound enzyme, phospholipase C (PLC)-b (Figure 1).

PLCb activation promotes the hydrolysis of phosphatidylinositol 4,5-bisphosphate (PIP_2_) into the second messengers diacylglycerol (DAG) and inositol 1′, 4′, 5′-trisphosphate (IP3), whose receptor is a Ca^2+^ channel in the endoplasmic reticulum (ER) membrane that, upon IP3 binding, releases Ca^2+^ from its ER stores into the cytoplasm, raising intracellular free [Ca^2+^] [50] (Figure 2). Increased intracellular free [Ca^2+^] serves as a second messenger in its own right. Of note, the free (i.e., dissociated from Gα) G_βγ_ subunits can also transduce signals and activate effectors. For instance, PLCβ is activated not only by Gαq subunits, but also by G_βγ_ subunits released from Gi/o proteins [51] (Figure 2).

AC, ion channels like the G protein-regulated inwardly rectifying K^+^ (GIRK) channels and voltage-gated Ca^2+^ channels, and phospho-inositide-3′-kinases (PI3Ks) are also effectors directly regulated by free G_βγ_ subunits [52]. G protein signaling is primarily terminated by three different protein families: the Regulator of G protein Signaling (RGS) proteins [53], which directly inactivate Ga subunits, the second messenger-dependent protein kinases, like protein kinase A (PKA) and protein kinase C (PKC), which phosphorylate and desensitize GPCRs, irrespective of their activity status [54], and GPCR-kinases (GRKs) acting in tandem with arrestins, which phosphorylate and desensitize (i.e., uncouple from the G protein) only agonist-activated GPCRs [55]. Notably, the β-arrestins (arrestin-2 and -3) can also internalize the phosphorylated receptor via clathrin-coated pits, thanks to their function as adaptors for clathrin adaptor protein (AP)-2 and clathrin itself [54]. In addition, as the receptor-β-arrestin complex traffics within endosomes, a second wave of G protein-independent signaling is initiated by the protein scaffolding actions of the β-arrestin [56]. This endosomal trafficking is also crucial for the resensitization of the GPCR, i.e., its recycling back to the membrane to signal again by a subsequent agonist activation [54]. Additional layers of GPCR signaling modulation are provided by the allosteric modulators of GPCRs, i.e., ligands that bind outside the agonist binding pocket of the receptor, and the dimerization/oligomerization of certain GPCRs, which is necessary for the signaling of certain GPCR types, such as the g-aminobutyric acid (GABA) type B (GABA-B) and the metabotropic glutamatergic (mGlu) receptors [57,58,59].

In the following sections, we discuss the signal transduction properties, pharmacology, and physiology/pathophysiology of the two SCFARs, first of FFAR2, and then of FFAR3, with an emphasis on their implications for the homeostasis and disorders of the cardiovascular system.

## 2. FFAR2 Signaling and Cardiovascular Function

### 2.1. Signaling of FFAR2

Although it has been suggested to be preferentially activated by shorter SCFAs, FFAR2 is activated by all three main SCFAs: acetate, propionate, and butyrate [60,61], albeit with species-dependent variation in agonist potency [62]. FFAR2 is known to couple to both Gi/o and G_q/11_ proteins [1,2] (Figure 1). Therefore, via Gi/o protein activation, it inhibits AC and lowers intracellular cAMP levels, but also activates the MAPKs ERK1/2 at the same time (Figure 1). On the other hand, via G_q/11_ protein activation, FFAR2 increases intracellular [Ca^2+^] and also promotes activation of ERKs and other MAPKs (Figure 1). Importantly, FFAR2 also activates β-arrestin2, thereby inhibiting the nuclear translocation and hence, the activation of the pro-inflammatory transcription factor nuclear factor (NF)-κB, which reduces the synthesis of pro-inflammatory cytokines, such as interleukin (IL)-1β and IL-6, in heterologous cells (Figure 1) [63].

### 2.2. Function of FFAR2 in Relation to the Cardiovascular System

The roles of FFAR2 in food allergies, cancer, arthritis/gout, and autoimmune disorders, including type 1 diabetes, have been well established [1]. Regarding its potential effects in the cardiovascular system, however, the current knowledge is, unfortunately, next to none. FFAR2 regulates the permeability of the blood-brain barrier (BBB) [64] and increases glucagon-like peptide (GLP)-1 and peptide YY (peptide tyrosine-tyrosine, PYY) synthesis [65]. FFAR2 agonist treatment induces PYY and GLP-1 secretions in mice via the Gi/o protein/AC inhibition signaling pathway in intestinal cells [66] (Figure 1). Although both of these hormones (and especially GLP-1) are known to have several beneficial actions for the heart [67,68], the extent (if any) to which FFAR2 affects cardiovascular function by regulating the production of GLP-1 and/or PYY is completely unknown at this time. In contrast, the effects of FFAR2 on GLP-1 and PYY levels shed light on the therapeutic value of FFAR2 pharmacological targeting for diabetes, obesity, and other metabolic disorders.

One of the most important functions of FFAR2 is the regulation of energy accumulation in adipose tissues and of adipogenesis, thus having significant ramifications for metabolic syndrome pathogenesis [69]. Indeed, FFAR2 has been shown to increase adipogenesis [70]. Acetate and propionate upregulate FFAR2 in murine fat tissues, leading to lower plasma FFA levels and decreased lipolysis [70,71]. The pro-adipogenic role of FFAR2 seems to be corroborated by studies in FFAR2 knockout mice fed a high fat diet (HFD), which then displayed lower body fat mass, improved glucose control, lower plasma FFA levels, increased energy expenditure and brown adipose tissue (BAT) density (“browning” of adipose tissue), as well as lower white adipose tissue (WAT) inflammation, suggesting FFAR2 as a crucial mediator in HFD-induced obesity/diabetes [28,72]. However, other studies have failed to show any effect of SCFAs on adipogenesis in vitro or in vivo, or any FFAR expression level alterations, refuting the correlation of FFAR2 with human adiposity [72,73]. Thus, the role of FFAR2 in human fat tissue homeostasis and development remains controversial and unclear at this point (Figure 1). If future studies prove a causative role for this receptor in human adipogenesis and obesity, then its pharmacological inhibition would be theoretically advantageous for heart disease, as well. A definitive answer to this conundrum however, is, at best, several years away.

## 3. FFAR3 Signaling and Cardiovascular Function

### 3.1. Signaling of FFAR3

FFAR3 was deorphanized in 2003 (it was called GPR41 until then), when it was identified as a SCFAR [60,74]. Similar to FFAR2, FFAR3 is activated by all the main SCFAs, such as propionate, butyrate, and valerate, all produced by the bacterial metabolism of otherwise indigestible dietary fiber in the gut [1]. However, in contrast to FFAR2, FFAR3 is minimally activated by acetate (the shortest FFA that exists in nature) and shows a preference for the longest-chain SCFAs (valerate with 5 C atoms, caproate with 6 C atoms) for activation [60,74]. Additionally, FFAR3 signaling seems to proceed exclusively via the pertussis toxin-sensitive Gi/o proteins (Figure 2), unlike FFAR2, which can couple to G_q/11_ proteins, as well. Indeed, FFAR3 stimulation with SCFAs inhibits AC and lowers intracellular cAMP synthesis via Gai subunit activation, but also promotes ERK1/2 phosphorylation and activation via Gi/o-derived free G_βγ_ subunits [75,76] (Figure 2). Of note, although FFAR3 is not known to couple to the G_q/11_ protein pathway, it can also induce the phosphoinositide hydrolysis cascade and stimulate intracellular Ca^2+^ signaling, like FFAR2 does, again via the Gi/o-derived free Gβγ subunit activation of PLCβ_2/3_ [21] (Figure 2).

### 3.2. Function of FFAR3 in Relation to the Cardiovascular System

Unlike FFAR2, whose role in cardiovascular homeostasis (if any), as mentioned above, is virtually unknown, FFAR3’s involvement in cardiovascular function regulation has become increasingly clear over the past decade. The main physiological mechanism by which FFAR3 regulates cardiac function is via the effects it exerts in the sympathetic nervous system [21]. FFAR3 is robustly expressed in murine peripheral sympathetic neurons, including cardiac sympathetic nerve terminals, wherein it regulates whole body metabolic homeostasis, along with neuronal activity/firing by inducing norepinephrine release [21] (Figure 2). Although both norepinephrine and epinephrine mediate the effects of the sympathetic nervous system on all cells and tissues of the entire body, norepinephrine is the actual neurotransmitter synthesized, stored, and released from sympathetic neurons [77,78,79]. Epinephrine is the hormone synthesized in the adrenal medulla and secreted into the systemic circulation [77,78,79]. This is because sympathetic neurons lack the enzyme phenyl-ethanolamine-N-methyltransferase (PNMT), which converts norepinephrine to epinephrine [78,80]. FFAR3 is also present in portal neurons of the liver, where it regulates propionate-induced gluconeogenesis via the gut-brain axis [81]. FFAR3 knockout mice display significantly lower catecholamine (norepinephrine and epinephrine) synthesis, as evidenced by the downregulation of tyrosine hydroxylase, the enzyme that catalyzes the rate-limiting step of catecholamine biosynthesis [21] (Figure 2). Consistent with a lower sympathetic neuronal activity/firing rate, heart rate is also reduced in FFAR3 knockout mice [21] (Figure 2). Thus, FFAR3 clearly promotes neuronal firing and norepinephrine synthesis and release in sympathetic neurons. The signaling pathway underlying this effect of FFAR3 is the stimulation of the Gi/o-derived free Gbg subunit activation of PLCβ_2/3_ (see above) [21]. G_βγ_-activated PLCβ_2/3_ activates, in turn, the MAPKs ERK1/2, which phosphorylate synapsin-2β at Ser426 to induce vesicle fusion with the neuronal plasma membrane and norepinephrine exocytosis/synaptic release from sympathetic nerve endings [82] (Figure 2). Notably, neither GRK2 nor β-arrestins appear to be involved in this signaling pathway [21], although GRK2 should theoretically play a role, since it interacts with free Gβγ subunits via its C-terminal pleckstrin homology (PH) domain [54]. In fact, this is the main mechanism for membrane targeting and the activation of GRK2 (and GRK3) [55]. On the other hand, RGS proteins of the B/R4 family, which inactivate Gi/o proteins, must interfere with this FFAR3-dependent signaling pathway in sympathetic neurons [53]. RGS4 in particular is known to directly bind Gi/o-derived free G_βγ_ subunits and PLCβ and to inhibit PLCβ activation independently of its RGS function [83,84,85]. Indeed, while studying FFAR3 signaling and function in rat cardiomyocytes, we have confirmed that RGS4 intervenes in FFAR3 signaling to PLCβ via the Gi/o-derived free G_βγ_ subunits, dampening the subsequent PLCβ-induced Ca^2+^ signaling from this receptor and leading to inflammation in the heart, as well as norepinephrine release and firing activity in cardiac sympathetic neurons [86] (Figure 2). Nevertheless, other studies have suggested that FFAR3 may actually inhibit secretion/exocytosis via N-type Ca^2+^ channel inhibition in enteric and vascular neurons [87,88]. More specifically, FFAR3 signaling inhibits N-type Ca^2+^ channels via G_βγ_ signaling, reducing neuronal catecholamine release in rat sympathetic neurons innervating vascular smooth muscle [87]. Additionally, FFAR3 modulates the cholinergic-mediated secretory response in the proximal colonic mucosal neurons of rats [88], and FFAR3 is a putative target for neurogenic bowel disorder treatment [89]. Indeed, the FFAR3 synthetic agonist AR420626 suppresses cholinergic and serotonergic-dependent colonic motility and secretions [89]. Therefore, the picture regarding neuronal FFAR3 effects is undoubtedly complicated, and more studies are required to provide better clarity. The decade-old study by Kimura et al. in FFAR3 knockout mice demonstrated that the FFAR3-dependent norepinephrine release from sympathetic neurons modulates energy expenditure and that the activation of FFAR3 with propionic acid elevates heart rate and increases cardiac oxygen demand/consumption [21]. It also showed that the effect of propionate/FFAR3 on heart rate was suppressed by pretreatment with a β-adrenergic receptor (AR) blocker, which indicated that FFAR3 signaling is reciprocally regulated by βARs, i.e., there is a signaling crosstalk between FFAR3 and βARs, upregulating βAR function in sympathetic ganglions [21]. Notably, in that same study, the ketone body β-hydroxybutyrate (or 3-hydroxybutyrate) was shown to block FFAR3 and antagonize its pro-sympathetic hyperactivity in neurons [21] (Figure 2). On the other hand, sodium/glucose co-transporter (SGLT)-2 inhibitors, such as dapagliflozin and empagliflozin (anti-diabetic/diuretic drugs with a plethora of beneficial cardiovascular effects that have been coming to light at an accelerating pace), increase ketone body (including β-hydroxybutyrate) production in the heart and blood vessels [90,91]. Given that dapagliflozin and empagliflozin have been shown to possess sympatholytic properties that mediate, at least in part, their beneficial effects in chronic human heart failure [92], it is tempting to speculate that the sympatholytic effects of SGLT2 inhibitor drugs are mediated, at least partially, by the b-hydroxybutyrate-mediated blockade of FFAR3 signaling in sympathetic neurons, which normally raises cardiac norepinephrine levels and cardiovascular sympathetic nervous system activity [91]. This, of course, awaits confirmation by future studies in experimental models of chronic heart failure.

FFAR3 is expressed, not only in postganglionic sympathetic and sensory neurons of the autonomic nervous system, but also in sympathetic and sensory neurons of the somatic peripheral nervous system [21,87,93]. Thus, SCFAs exert their effects via FFAR3 not only through the enteroendocrine system, but also directly by modifying physiological reflexes integrating the peripheral nervous system and the gastrointestinal tract. Moreover, FFAR3 in submucosal, and the myenteric ganglionic plexus neurons of the small intestine regulate gut hormonal synthesis, including GLP1 and PYY synthesis, similar to FFAR2 (see above) [3,5,11]. Additionally, FFAR3 significantly reduces lipolysis by inhibiting hormone-sensitive lipase phosphorylation and activity via Gαi_-_mediated AC inhibition and cAMP lowering in peripheral adipose tissues [15,94] and increases leptin production, hepatic lipogenesis, and adipocyte growth [4] (Figure 2). Indeed, male FFAR3 knockout mice treated with HFD exhibit more body fat mass and higher blood glucose levels compared to wild type female littermates [95], and leptin synthesis is reduced in FFAR3 knockouts [96]. Furthermore, porcine FFAR3 activated by butyrate augments lipid accumulation and adipogenesis via Akt (protein kinase B) and 5′-adenosine monophosphate-activated kinase (AMPK) signaling [97] (Figure 2). Thus, FFAR3 induces satiety through gut-brain hormonal axis regulation [16,20], thereby controlling total body energy metabolism. In addition, propionate exerts protective effects on the BBB via FFAR3 activation on the surface of endothelial cells [98], suggesting another role for FFAR3 in mediating the effects of gut-derived microbial SCFA metabolites in the modulation of the gut-brain axis. Nevertheless, as stated above for FFAR2, the manner and extent to which the metabolic effects of FFAR3 and its role in gut-brain axis maintenance/homeostasis influence this receptor’s effects on cardiovascular regulation are completely elusive at present.

In contrast, a study on Olfr78, an olfactory receptor expressed in the juxtaglomerular apparatus of the kidney and activated by SCFAs, and on FFAR3, demonstrated a clear role for vascular FFAR3 in the regulation of blood pressure/vascular tone and in hypertension [18] (Figure 2). More specifically, Olfr78 and FFAR3 were found to be expressed in the smooth muscle cells of small resistance blood vessels, and Olfr78 knockout mice, as well as FFAR3 knockout mice, developed hypertension upon antibiotic treatment, which reduced SCFA levels derived from gut microbial fermentation [18]. These findings were consistent with older studies showing that propionate and other SCFAs induce vasodilation ex vivo, thereby producing acute hypotensive responses, and are associated with anti-hypertensive protection [99,100,101,102]. However, the signaling mechanism(s) underlying this hypotensive effect of FFAR3 were not investigated. On the other hand, the FFAR3 knockout mice studied were global knockouts, i.e., lacked FFAR3 expression in all cells/tissues, so whether this is an effect of FFAR3 mediated by the endothelium or smooth muscle of the resistance vessels (or both) is not known. Further complicating the enigma of the underlying mechanism(s) is the fact that FFAR3 signals through Gi/o proteins, which a) lower levels of cAMP, a second messenger that induces the relaxation of vascular smooth muscle (vasodilatory), and b) induce PLCβ-Ca^2+^ signaling via G_βγ_ subunits (see above), which also normally induces the contraction of vascular smooth muscle (vasoconstriction). Moreover, the fact that FFAR3 promotes norepinephrine release from sympathetic nerve terminals (see above, [21]), including those that innervate the renal juxtaglomerular apparatus inducing renin release via β_1_ARs [103,104], should also result in the elevation, rather than reduction, of blood pressure by FFAR3. Perhaps then, the vasodilatory effect of propionate-activated FFAR3 is mediated by endothelial nitric oxide production secondary to Ca^2+^-dependent endothelial NO synthase (eNOS) activation [105]. In any case, the precise mechanism(s) underlying the reported anti-hypertensive, vasodilating effects of FFAR3 warrants elucidation in future studies.

## 4. Concluding Remarks

SCFAs acting through FFAR2 and -3 regulate a vast variety of biological processes, such as energy metabolism, adipogenesis, appetite control, intestinal cellular homeostasis, gut motility, glucose metabolism, inflammation, and central and autonomic (sympathetic) nervous system function. It is thus no surprise that perturbations in signaling and function of these two SCFARs cause or contribute to several human disorders, including, but not limited to, diabetes, obesity, gout, arthritis, colitis, and asthma, as well as cardiovascular diseases, such as hypertension, atherosclerosis, cardiac arrhythmias, and heart failure. Although a great deal about the physiology and biology of these receptors still awaits elucidation, particularly with regard to their roles in cardiovascular homeostasis, it appears that FFAR3 has a far larger and clearer role in cardiovascular regulation than does FFAR2, courtesy of its prominent effects on sympathetic neurons and norepinephrine release. Due to its well documented pro-catecholaminergic effects, it seems that the pharmacological inhibition of FFAR3 with a synthetic FFAR3-selective antagonist or with ketogenic drugs (e.g., SGLT2 inhibitors) that increase levels of ketone bodies (e.g., 3-hydroxybutyrate), FFAs that act as natural, endogenous FFAR3 antagonists, has the potential to treat several cardiovascular diseases aggravated by sympathetic nervous system hyperactivity, such as chronic heart failure, hypertension, coronary artery disease, atrial fibrillation, etc. However, there are some caveats to this premise. For example, despite the mystery surrounding the underlying mechanism, vascular FFAR3 appears to promote vasodilatation, a therapeutically desirable effect in heart disease patients. Another interesting twist comes from a biophysical study that revealed that FFAR2 and FFAR3 heterodimerize with each other, and the dimer’s signaling properties are distinct from those of the constituting protomers [106]. Specifically, the FFAR2/FFAR3 heterodimer displays increased intracellular Ca^2+^ signaling vs. monomeric FFAR2 and robustly enhanced β-arrestin2 recruitment vs. monomeric FFAR3, along with a lack of AC inhibition [106]. Since this study was performed in primary human monocytes and macrophages, as well as in transfected human embryonic kidney (HEK)-293 cells, it is not known whether this FFAR2/FFAR3 heterodimerization actually occurs in vivo and consequently, what its physiological relevance (if any) is. Nevertheless, the possibility of FFAR2/FFAR3 heterodimers in vivo is another layer of complexity that needs to be considered for SCFAR-targeted drug development.

Although much work remains to be done, a plethora of in vitro and in vivo studies have already uncovered the physiological functions of SCFARs in the regulation of body metabolism, energy utilization, and immune system function/inflammation. With respect to cardiovascular homeostasis modulation, unfortunately, very little is known about the roles these receptors play. The overlapping signaling modalities, tissue expression patterns, and functions of FFAR2 and FFAR3 present immense obstacles in the research efforts to delineate their physiology and pharmacology. This apparent functional redundancy of the SCFARs (and of all four different FFARs, for that matter) likely signifies that their roles in organ system homeostasis, including cardiovascular homeostasis, are modulatory and adjuvant, rather than causal or absolutely essential. Nevertheless, FFAR2 and FFAR3 appear to play important roles in modulating biological processes in response to nutritional state changes and in linking dietary effects with cardiovascular function or disease. Future studies on the role of SCFARs in cardiovascular biology and heart diseases will be instrumental in determining the precise contributions of diet and of gut microbiota in cardiovascular pathologies. This will enable the development and utilization of SCFAR-targeting drugs in clinical practice, not only for the treatment of diabetes, obesity, metabolic syndromes, and of inflammatory disorders, but also for use by the cardiologists of the future.

## Figures and Tables

**Figure 1 ijms-23-03303-f001:**
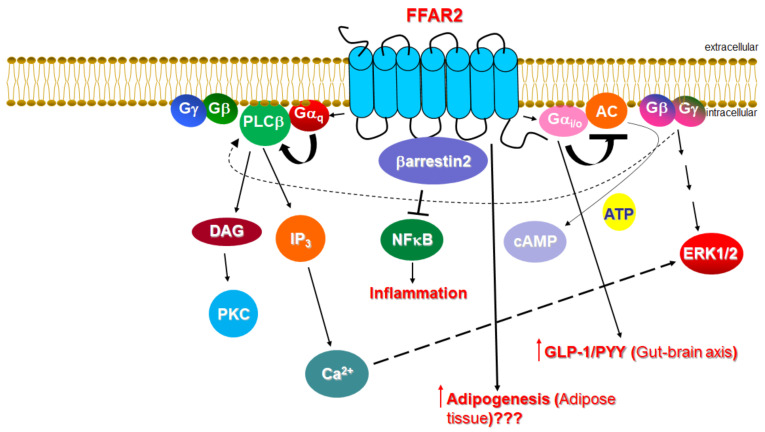
The cardiovascular physiology relevant to FFAR2 signaling. AC: Adenylyl cyclase; DAG: Diacylglycerol; ERK: Extracellular signal-regulated (mitogen-activated protein, MAP) kinase; IP_3_: Inositol 1′, 4′, 5′-trisphosphate; PLC: Phospholipase C; PKC: Protein kinase C; “???” indicates a lack of consensus (currently) for the action depicted. (See text for details and for all other molecular acronym descriptions.)

**Figure 2 ijms-23-03303-f002:**
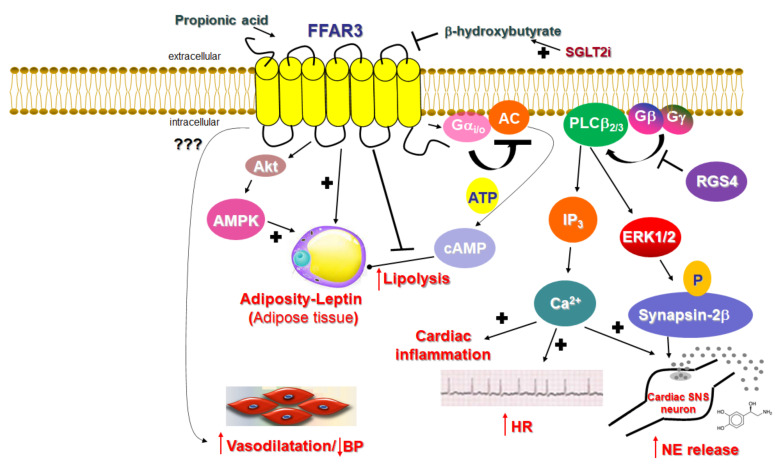
The cardiovascular physiology relevant to FFAR3 signaling. BP: Blood pressure; ERK: Extracellular signal-regulated (mitogen-activated protein, MAP) kinase; HR: Heart rate; IP_3_: Inositol 1′, 4′, 5′-trisphosphate; NE: Norepinephrine (noradrenaline); P: Phosphorylation; RGS4: Regulator of G protein signaling protein-4; SGLT2i: Sodium-glucose co-transporter type 2 inhibitor; SNS: Sympathetic nervous system; “???” indicates a signaling mechanism that is (currently) unknown. (See text for details and for all other molecular acronym descriptions.)

## Data Availability

Not applicable.

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
