# Peer review of "Short-Chain Fatty Acid Receptors and Cardiovascular Function"

_ijms, 2022, doi:10.3390/ijms23063303_

Round 1

Reviewer 1 Report

The manuscript presented consider an aspect but I think that at the moment it is just ana hypotesys 

Line 119 The implication is still a hypothesis, as a cause-and-effect relationship is not well established

The whole paragraph of FFAR2 is however on animal models, which especially as regards the adipose tissue differs a lot from what happens to human

I think that the whole work is quite speculative since we assume the influence of SCFAs in various pathologies that have a complex etiology and with well-known causes; for example, type II diabetes is clearly caused by a lack of physical activity and incorrect nutrition, there is no need to consider anything else ..... the same goes for obesity, even the colitis that is mentioned is only a symptom, probably of food intolerances, which in turn, by altering the microbiota, induce the synthesis of SCFAs, therefore the latter is a consequence, not a cause.

Even more, the correlation with cardiovascular function is hypothetical and the chance to be a therapeutic target seems very far

Author Response

1) The manuscript presented consider an aspect but I think that at the moment it is just ana hypotesys 

Author response: We thank this reviewer for the overall kind and positive comments about the quality of our work. However, we are not sure what he/she means here. Our present manuscript is a literature review article, not an original research or research hypothesis study.

2) Line 119 The implication is still a hypothesis, as a cause-and-effect relationship is not well established

Author response: Thank you for your comment but we respectfully disagree. The roles of FFAR2 in cancer, autoimmune diseases, arthritis, etc., are well established (e.g., see Refs. #1 & #2 of our manuscript for excellent, recent reviews on this topic).

3) The whole paragraph of FFAR2 is however on animal models, which especially as regards the adipose tissue differs a lot from what happens to human

Author response: We agree with the reviewer on this pertinent remark. Unfortunately, not much is known about the effects of human FFAR2 in human adipose tissues.  

4) I think that the whole work is quite speculative since we assume the influence of SCFAs in various pathologies that have a complex etiology and with well-known causes; for example, type II diabetes is clearly caused by a lack of physical activity and incorrect nutrition, there is no need to consider anything else ..... the same goes for obesity, even the colitis that is mentioned is only a symptom, probably of food intolerances, which in turn, by altering the microbiota, induce the synthesis of SCFAs, therefore the latter is a consequence, not a cause.

Author response: Although we agree with the reviewer that these pathologies are complex, we do not understand the comment about “the whole work is quite speculative”. This is a review discussing the current literature on FFAR2 and FFAR3 in relation to the cardiovascular system, not a research study. Besides, we never claimed that SCFAs are causing these pathologies, since that would be untrue. SCFAs simply modulate the pathophysiology of these diseases and that`s precisely what we discuss and describe in our review manuscript. Finally, we respectfully disagree with the comment about type 2 diabetes: adiposity, immune dysfunction, and inflammation play major parts in its pathophysiology; malnutrition and poor physical activity are only two of the several causative factors of this disease. The same goes for obesity, as well. We hope that the reviewer concurs with us on these scientific facts.

5) Even more, the correlation with cardiovascular function is hypothetical and the chance to be a therapeutic target seems very far

Author response: As we clearly state at the end of Section 2.2 and in “Concluding Remarks” (Section 4), FFAR2`s correlation with cardiovascular function is indeed weak and rather speculative at present. In contrast, that of FFAR3 is quite clear and well established via numerous studies that we have cited and discussed in our present review manuscript.

Reviewer 2 Report

The authors focused on the effects of short fatty acid receptors on cardiovascular disease.These receptors are quite common in many tissues, including the heart . Authors discussed functions of this receptors in regulating neurohormonal control of the circulation and adipose tissue homeostasis. Authors also emphasize the use of receptors as a therapeutic target.

The advantage of this review are well-made figures summarizing the subject matter.

Next advantage is that the authors used the new literature on these issues.

As to my comments:

I would add an explanation of the AC, PLC abbreviations to the description of Figure 1.

In the last sentence of section 3.1 different sizes of fonts are used, the same in section 3.2 verse 189, 202, 205, 274, 281.

Could you explain abbreviations: HFD in verse 245, BBB verse 252.

Author Response

1) The authors focused on the effects of short fatty acid receptors on cardiovascular disease.These receptors are quite common in many tissues, including the heart . Authors discussed functions of this receptors in regulating neurohormonal control of the circulation and adipose tissue homeostasis. Authors also emphasize the use of receptors as a therapeutic target.

The advantage of this review are well-made figures summarizing the subject matter.

Next advantage is that the authors used the new literature on these issues.

Author response: We thank this reviewer for the very kind and positive comments about the quality of our work.

As to my comments:

2) I would add an explanation of the AC, PLC abbreviations to the description of Figure 1.

Author response: Done, thank you.

3) In the last sentence of section 3.1 different sizes of fonts are used, the same in section 3.2 verse 189, 202, 205, 274, 281.

Author response: Font size corrected, thank you.

4) Could you explain abbreviations: HFD in verse 245, BBB verse 252.

Author response: Explained in lines 137 & 122, thanks again.

Round 2

Reviewer 1 Report

Author response: We thank this reviewer for the overall kind and positive comments about the quality of our work. However, we are not sure what he/she means here. Our present manuscript is a literature review article, not an original research or research hypothesis study.

2) Line 119 The implication is still a hypothesis, as a cause-and-effect relationship is not well established

Author response: Thank you for your comment but we respectfully disagree. The roles of FFAR2 in cancer, autoimmune diseases, arthritis, etc., are well established (e.g., see Refs. #1 & #2 of our manuscript for excellent, recent reviews on this topic).

well established means that there is a clear mechanism, verified in humans, reproducible and present in the aforementioned pathologies and this is not the case

3) The whole paragraph of FFAR2 is however on animal models, which especially as regards the adipose tissue differs a lot from what happens to human

Author response: We agree with the reviewer on this pertinent remark. Unfortunately, not much is known about the effects of human FFAR2 in human adipose tissues.

Therefore it should be well underlined that everything that is proposed is not on man, for this reason, it is a hypothesis that it can work in the same way on him

4) I think that the whole work is quite speculative since we assume the influence of SCFAs in various pathologies that have a complex etiology and with well-known causes; for example, type II diabetes is clearly caused by a lack of physical activity and incorrect nutrition, there is no need to consider anything else ..... the same goes for obesity, even the colitis that is mentioned is only a symptom, probably of food intolerances, which in turn, by altering the microbiota, induce the synthesis of SCFAs, therefore the latter is a consequence, not a cause.

Author response: Although we agree with the reviewer that these pathologies are complex, we do not understand the comment about “the whole work is quite speculative”. This is a review discussing the current literature on FFAR2 and FFAR3 in relation to the cardiovascular system, not a research study. Besides, we never claimed that SCFAs are causing these pathologies, since that would be untrue. SCFAs simply modulate the pathophysiology of these diseases and that`s precisely what we discuss and describe in our review manuscript. Finally, we respectfully disagree with the comment about type 2 diabetes: adiposity, immune dysfunction, and inflammation play major parts in its pathophysiology; malnutrition and poor physical activity are only two of the several causative factors of this disease. The same goes for obesity, as well. We hope that the reviewer concurs with us on these scientific facts.

No, I absolutely disagree, treating hundreds of obese and diabetic patients, I can say with certainty that the main causes and therefore the treatment is incorrect nutrition and lack of physical activity, there is no need to investigate particular molecular mechanisms, which can compete but not to be the cause.

What exactly does modulating the pathology mean? If a mechanism can modulate a pathology or is also the cause or is induced, but by what?

5) Even more, the correlation with cardiovascular function is hypothetical and the chance to be a therapeutic target seems very far

Author response: As we clearly state at the end of Section 2.2 and in “Concluding Remarks” (Section 4), FFAR2`s correlation with cardiovascular function is indeed weak and rather speculative at present. In contrast, that of FFAR3 is quite clear and well established via numerous studies that we have cited and discussed in our present review manuscript.

Author Response

well established means that there is a clear mechanism, verified in humans, reproducible and present in the aforementioned pathologies and this is not the case

Author reply: We agree this is not the case for FFAR2 in adipogenesis & cardiovascular function. That`s why we clearly state throughout Section 2.2 that  FFAR2`s connection with adipogenesis and obesity is theoretical at this point and more studies are needed. We hope this satisfies this reviewer.

Therefore it should be well underlined that everything that is proposed is not on man, for this reason, it is a hypothesis that it can work in the same way on him

Author reply: We have difficulty understanding this comment. Whom is the reviewer referring to by "him", a specific person or patient? Anyway, as stated in our reply above, once again, we clearly mention throughout Section 2.2  of our manuscript that FFAR2`s connection with human adipogenesis and obesity is hypothetical at present and more studies need to be done in the future to fully delineate this.  

No, I absolutely disagree, treating hundreds of obese and diabetic patients, I can say with certainty that the main causes and therefore the treatment is incorrect nutrition and lack of physical activity, there is no need to investigate particular molecular mechanisms, which can compete but not to be the cause.

Author reply: We apologize but, once again, we do not understand what exactly the reviewer means here. Is the reviewer arguing that obesity and diabetes can only be treated with good nutrition and physical activity? In other words, medications have NO place in treatment and NO pharmacological mechanisms are involved??? Without getting into the substance of this argument, we would like to point out that it is irrelevant to our present review, anyway, since our present article only deals with the cardiovascular effects of FFAR2 & FFAR3, not with their effects in obesity or diabetes.  

What exactly does modulating the pathology mean? If a mechanism can modulate a pathology or is also the cause or is induced, but by what?

Author reply: It simply means that a factor (SCFA and its receptor in this case) is NOT the cause of the disease but its effects can aggravate (or ameliorate) the overall pathology/symptomatology of the disease (e.g., morbidity, mortality, etc.). For instance, sympathetic hyperactivity does NOT cause chronic heart failure but it significantly increases the morbidity and mortality of this disease.